# Challenges with Regard to Unmanned Aerial Systems (UASs) Measurement of River Surface Velocity Using Doppler Radar

**Filippo Bandini ***, **Monica Coppo Frías, Jun Liu, Kasparas Simkus, Sofia Karagkiolidou and Peter Bauer-Gottwein**

Department of Environmental Engineering, Technical University of Denmark, 2800 Kongens Lyngby, Denmark; mcfr@env.dtu.dk (M.C.F.); juli@env.dtu.dk (J.L.); s180210@student.dtu.dk (K.S.); s182574@student.dtu.dk (S.K.); pbau@env.dtu.dk (P.B.-G.)
* Correspondence: fban@env.dtu.dk

**Abstract:** Surface velocity is traditionally measured with in situ techniques such as velocity probes (in shallow rivers) or Acoustic Doppler Current Profilers (in deeper water). In the last years, researchers have developed remote sensing techniques, both optical (e.g., image-based velocimetry techniques) and microwave (e.g., Doppler radar). These techniques can be deployed from Unmanned Aerial Systems (UAS), which ensure fast and low-cost surveys also in remotely-accessible locations. We compare the results obtained with a UAS-borne Doppler radar and UAS-borne Particle Image Velocimetry (PIV) in different rivers, which presented different hydraulic–morphological conditions (width, slope, surface roughness and sediment material). The Doppler radar was a commercial 24 GHz instrument, developed for static deployment, adapted for UAS integration. PIV was applied with natural seeding (e.g., foam, debris) when possible, or with artificial seeding (woodchips) in the stream where the density of natural particles was insufficient. PIV reconstructed the velocity profile with high accuracy typically in the order of a few cm s$^{-1}$ and a coefficient of determination ($R^2$) typically larger than 0.7 (in half of the cases larger than 0.85), when compared with acoustic Doppler current profiler (ADCP) or velocity probe, in all investigated rivers. However, UAS-borne Doppler radar measurements show low reliability because of UAS vibrations, large instrument sampling footprint, large required sampling time and difficult-to-interpret quality indicators suggesting that additional research is needed to measure surface velocity from UAS-borne Doppler radar.

**Keywords:** Unmanned Aerial Systems (UAS); drone; Doppler radar; Particle Image Velocimetry (PIV); river; velocity

## 1. Introduction

Measurements of surface velocity in rivers are essential to evaluate sediment deposition and erosion processes [1,2] especially during high flow periods [3], study fish migration [4–6] and estimate discharge [7–12]. In shallow rivers, where the operator can wade through, velocity is generally measured with a velocity probe (propeller-based or electromagnetic instruments), whilst Acoustic Doppler Current Profiler (ADCP) is typically used in rivers with depths larger than 1–2 m. Researchers have also experimented with contactless techniques for measuring surface velocity, such as optical and radar methods. Among the optical methods, Particle Image Velocitmetry (PIV) and Particle Tracking Velocimetry (PTV), are the most widely used methods [13–17], although recently, new algorithms have been developed and cross-compared (e.g., [18,19]). The PTV technique is based on the identification and tracking of individual particles [20–23], whilst PIV recognizes and tracks patterns (which can be a group of particles in an interrogation area on the water surface) [24–27]. Many studies have demonstrated the application of tracking techniques to videos acquired from Unmanned Aerial Systems (UASs) for surface velocity estimation: e.g., Tauro et al. in [28,29] deployed UAS-borne PIV to estimate surface velocity maps and

compared them with traditional velocity measurements (flowmeters) and stationary PIV deployment; Detert et al. [30] used PIV to analyse videos retrieved from a low-cost UAS and compared velocity estimates with ADCP; Koutalakis et al. [15] compared UAS-borne application of PIV, PTV and space-time image velocimetry (STIV); Thumser et al. [14] tested a new river surface velocity technique based on PTV and developed a camera able to track infrared-light emitting tracer particles; Bandini et al. [7] used UAS-borne PIV to estimate surface velocity, Manning's number and discharge.

Radar velocimetry methods are typically based on the Doppler effect, i.e., the change in frequency of a signal reflected by a moving target. In particular, when microwaves are incidental on rough water surfaces at intermediate incidence angles, they are scattered back to the receiver by short surface waves [31]. This phenomenon is known as composite surface scattering in which Bragg-resonant scattering is predominant. Bragg scattering is mainly caused by short surface waves (length is defined by the Bragg resonance condition, typically a few centimeters) [32–35]. The short waves causing Bragg-resonant scattering are generally caused by wind, or indirectly by longer waves, by the turbulence of the flow, or by rainfall [36,37].

Plant et al. [31] developed and tested a continuous wave Doppler microwave system (24 GHz), a pulsed Doppler radar (10 GHz), and an airborne real aperture coherent radar (9.36 GHz). Measurements of river surface velocity from bridges, cableways, and river banks in rivers (with maximum velocity > 2 m s$^{-1}$) were shown to be accurate to within ca. 10 cm s$^{-1}$ when compared with in situ measurements for both continuous wave and pulse radar; however, the authors reported that water roughness is sometimes insufficient to ensure sufficiently strong return signals along the entire cross-section (XS) of natural streams. Furthermore, Plant et al. [31] (the results of the described test are also reported in [38]) also tested helicopter-borne and airplane-borne deployment of a real aperture coherent radar (a slightly modified version of the pulse radar). In this case, the transmitting and receiving antennas were separated so that signals from shorter ranges could be received without interference from ringing of the transmitted pulse. The helicopter was flown perpendicularly to the flow with a set of transmitting/receiving antennas mounted on each side of the helicopter to be able to cancel out the velocity component caused by the helicopter-generated downwash. The helicopter surveys showed that velocity can be recorded and compared with ground-truth if the helicopter is flown only a few meters above the water surface and if the propeller-generated downwash is used to increase water roughness, whilst the airplane test was less successful and showed that velocities could be detected intermittently only and ground truth comparison could not be performed. Furthermore, both tests showed that the main flow direction should be in the antenna look direction in order to obtain reliable results.

Plant et al. [31] also examined the possibility to detect surface velocity with a space-borne or air-borne along-track interferometric synthetic aperture radar (InSAR) and its limitation in terms of spatial resolution, minimum river width, water roughness and return power. Recently, Biondi at al. [39] have demonstrated the possibility to measure surface velocity with UAVSAR, an L-band SAR (developed by the National Aeronautics and Space Administration (NASA)), the characteristics of which are reported in [40], showing that high velocity measurements correlated well with the river portions where high velocities are expected from river morphology.

However, although handheld and stationary (e.g., bridge-mounted) radars proved able to monitor surface velocity [41–44] (but were shown to be less reliable in low flow conditions [45]), deployment of radar on UAS for measuring river surface velocity is a new and challenging development. Alimenti et al. [46] developed a prototype of a low-cost continuous wave (24 GHz) Doppler radar sensor that was deployed stationary from a bridge in two sites along the Tiber River (Italy) and provided promising results when compared to another reference radar and prior information of surface velocity distributions. In their outlook on future work, the authors discussed UAS deployment of this instrument. To date, there is only one study reporting actual UAS-deployment of a Doppler radar

sensor [47], in which Fulton et al. (2020) tested a continuous wave 24 GHz Doppler radar (developed by the United States Geological Survey and Sommer Messtechnik (Koblach, Austria)) in five flights over four different rivers in the United States of America (USA). UAS-borne observations were compared with another reference handheld radar and an acoustic Doppler velocimeter (ADV): differences were typically of a few cm s$^{-1}$ (ca. 1%) when comparing the surface velocity value retrieved at the location of maximum velocity in the couple of sites where the comparison between surface velocity with UAS-borne Doppler radar and ADV was shown; however, comparisons of the horizontal velocity profile across the XS are not reported in the paper. Furthermore, the authors reported that the radar was not working in sites with surface velocities less than 0.15 m s$^{-1}$ and that more research is needed to understand the effects of (i) the propeller-generated downwash on water velocity and roughness, (ii) sampling duration, (iii) quality of the scattered signal, and (iv) wind, in order to improve filtering methods and to evaluate performance in different hydrological and river morphological conditions.

In this paper, we report the application of a commercial Doppler radar for measuring water surface velocity in rivers with different hydraulic and morphological conditions (velocity, water roughness, slope, width). A commercial Doppler radar was readapted for UAS-borne use to ensure a faster survey time and was flown across the entire XS: indeed, a profile of surface velocity across the XS, instead of only measurements in one single location (e.g., where the maximum velocity occurs or in the center of the stream), is expected to provide better information about river flow. The paper shows the challenges with regard to using Doppler radar from a vibrating and unstable platform (UAS) and suggests that additional research on UAS-borne Doppler radars is needed to correct for moving platform effects and to better evaluate the quality of the water surface velocity signal. The Doppler radar was compared with UAS-borne PIV and with ground-truth (measured with electromagnetic velocity probe). PIV was chosen for this research because, whilst PTV may provide better results in some conditions (e.g., non-uniform or poor seeding density, unsteady flow), PTV techniques would also require particles of well-defined shape and with a particle size smaller than frame-to-frame displacement [17]. Thus, PIV was considered more suitable because natural seeding (e.g., foam or natural particles on the water surface) was used in most cases.

## 2. Materials and Methods

### 2.1. UAS Platform and Payload

In this research, the deployed drone was a DJI Matrice 600 PRO (DJI, Shenzhen, China). According to DJI, this hexacopter typically has a hovering accuracy of $\pm0.5$ m vertically and $\pm1.5$ m horizontally when Global Positioning System (GPS) position is used to stabilize the platform. We developed a UAS payload consisting of (i) the GNSS (Global Navigation Satellite System) NovAtel OEM7700 (NovAtel, Calgary, Alberta, Canada) receiver connected to Antcom (3G0XX16A4-XT-1-4-Cert) multiband antenna, (ii) the IWR1443BOOST radar from Texas Instrument (Dallas, Texas, USA), (iii) the RGB video-camera GoPro Hero 5 (GoPro, USA), (iv) the Doppler radar OTT SVR (OTT HydroMet, Kempten, Germany) and (v) the single-board computer BeagleBone Black (BeagleBoard.org), which logged and synchronized observations of the different sensors. Both the RGB camera and the IWR1443BOOST were mounted on the gimbal Gremsy T1 (Gremsy Co., Ltd., Ho Chi Minh City, Vietnam).

In this research, the IWR1443BOOST is used only to measure the range to water surface [48] and thus to ensure that, during the surface velocity observations, the UAS was kept at a precise elevation above the water surface. Thus, the radar is not used in real-time to control UAS altitude during hovering, but in post-processing of Doppler radar measurements to retain only the observations within the accepted range of the hovering altitude. The video-camera GoPro Hero 5 is used to retrieve a video sequence of the water flow in 1080p HD at 60 Hz.

The OTT SVR100 (from here on abbreviated as SVR (Surface Velocity Radar)) is a commercial 24 GHz (K-band) Surface Velocity Radar emitting radar pulses and detecting Doppler shift of the reflected pulses for water surface velocity estimation. It is specifically developed to be used in static deployment (i.e., typically mounted on bridges). The SVR uses Kalman filters with physical modeling of the water flow to give stable measurements even in turbulent conditions. The radar beam, with a measuring angle of 12° (Azimuth) and 24° (Elevation), covers an elliptical area on the water surface; however, the radar outputs only the averaged surface velocity in the illuminated area. The SVR should be mounted with a tilt between 30° and 60° with respect to the horizontal, with 30° resulting in the largest footprint but typically the strongest signal. For example, if the SVR was elevated 6 m above water surface with a tilt angle of 45°, it would result in an ellipsoidal footprint centered 6 m away from the radar, a major axis (streamwise) of 5.3 m and a minor axis (crosswise) of 1.8 m. The manufacturer datasheet states that the SVR can measure water velocity (range between 0.08 and 15 m s$^{-1}$) with a resolution of 0.1 mm s$^{-1}$ and accuracy of ±2% of the measured value. It can measure velocity from a distance between 0.5 to 20–25 m above the water surface. A minimum surface roughness is required for sufficient signal-to-noise ratio and the manufacturer suggests at least 1 mm minimum wave height.

The SVR is produced with a large case and heavy mounting brackets (which are used to set different mounting angles). The system was readapted to make it suitable for UAS deployment, specifically: (i) the case and mounting brackets were removed and substituted with a lightweight case, (ii) a new mounting bracket was designed to allow the user to set 3 different sensor tilt angles, (iii) vibration damping was provided to decrease the effect of high-frequency vibrations caused by the drone. The UAS-adapted version is shown in Figure 1a.

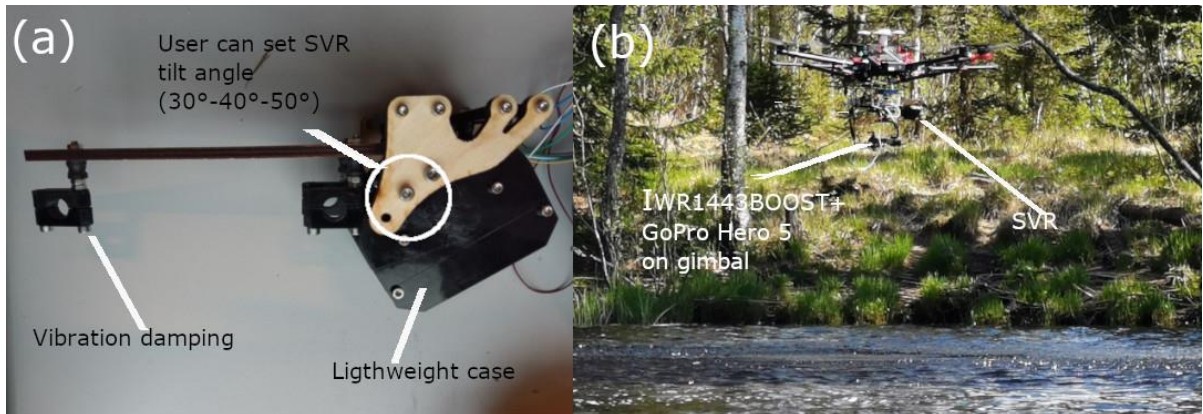

**Figure 1.** Pictures of UAS payload. (**a**) SVR adaption for being used as drone payload (**b**) UAS flight with SVR Doppler radar, IWR1443BOOST radar and GoPro Hero 5 RGB camera.

### 2.1.1. SVR Outputs

The SVR has different outputs (via the RS232 protocol): flow velocity, flow direction, mounting angle, vibration index, Signal-to-Noise Ratio (SNR), programmable gain amplifier (PGA) and another quality indicator of the detected level of the signal reflected from the water surface.

The SVR takes approximately 20 internal measurements per second; however, those high frequency measurements are not available to the user and are internally processed in the built-in processor of the SVR board. Indeed, SVR instantaneous measurements can be very scattered depending on factors such as roughness, wind influence and water turbulence. For this reason, the built-in processor of the SVR board applies filters to smooth out velocity measurements; specifically, two filters are available: infinite impulse response and moving average. The velocity value that is output via the RS232 protocol ca. once per second is internally computed after applying one of these two filters to the instantaneous velocity measurements. We chose to apply the moving average filter with a filter length

of 50 measurements, following the recommendations from the manufacturer. The flow direction can assume two values: 1 for approaching (when the sensor points upstream) and −1 for receding. The SVR tilt angle is measured with an inclinometer in degrees. The vibration index provides qualitative information and varies in a range from 0 (no vibrations, i.e., ideal measurement) to 3 (very strong vibrations, i.e., data unacceptable). SNR is an indication of the quality of the received signal (and is measured by superimposing the received signal to an interfering signal); it is defined as the logarithm of the ratio of the average power of the desired signal and the average noise power of the interfering signal. When larger energy is reflected from the water surface (typically for higher water surface roughness values), the overall signal strength is higher and, assuming constant noise, SNR should increase: SNR should be >6 decibel-milliWatts (dBm) for excellent signal quality. To improve SNR internally, the radar uses low-noise PGA. If the strength of the reflected signal is low, the radar will increase the gain level on PGA. The automatic gain control of the SVR automatically adjusts the PGA gain level. PGA is also an output of the SVR and, according to the manufacturer, should fall between 5 and 100 for ideal measurements: a PGA lower than 5 indicates very strong signal but may also indicate receiver saturation and thus reduced accuracy. The output SNR is available only in the latest firmware version, whilst the former firmware provided another indicator, the detected level of signal reflection, that was often hard to interpret in practice.

### 2.1.2. Ground-Truth Velocity Observations

In most cases, ground-truth observations were retrieved with the electromagnetic flowmeter OTT MF pro (OTT HydroMet, Kempten, Germany), from here on abbreviated as MFpro. The probe needs to be fully submerged in the flow, so observations could not be acquired exactly at the surface level but a few centimetres below the water surface [7,49].

In one XS, ground-truth observations were retrieved with the ADCP Teledyne Stream-Pro (Teledyne RD Instruments, Thousand Oaks, California, USA) instead of the MFpro. The ADCP has a blind zone at the surface level: surface velocity was thus assumed equal to the velocity measured at the depth closest to the surface (i.e., with the deployed ADCP typically 0.2 m below surface). During most of the surveys, less than one hour occurred between the end of the UAS measurements and the start of ground-truth measurements. Ground-truth measurements required ca. 0.5 h for ADCP and ca. 1 h for OTT MF pro.

### 2.2. Study Areas

Careful site selection is important to evaluate the performance of the UAV-borne SVR. Indeed, for retrieving at least 8–10 velocity points across the river, the footprint of the sensor required rivers with visible surface waves, whilst the minimum surface roughness requirement required a width of at least 10–15 m. Danish streams are representative of lowland streams at medium-high latitudes: they are characterized by mild slopes (in the order of a few cm or tens of cm per km) with clay or silt as riverbed material and dense aquatic vegetation during summer. For these reasons, stream velocity is typically rather low and water roughness is mainly caused by wind. In previous surveys (not reported in this paper), XSs in other Danish streams (Grindsted Å, Vejle Å) were surveyed with SVR but the surface roughness was insufficient for reliable measurements. Thus, surveys in Denmark were conducted in Guden Å, which is the largest river in Denmark in terms of discharge and river width. Furthermore, surveys were conducted in Swedish streams, which are typically larger, steeper and with stony or gravelly riverbeds, thus typically have higher water surface roughness than Danish rivers. In Sweden, three different XSs in three rivers (Krokån, Kilan and Ätran) were surveyed.

Table 1 shows the site names, the coordinates, the average depth, the stream width, the bed type and which velocity measurements were conducted. A map with the location of the sites is shown in Figure 2.

**Table 1.** Overview of the surveyed sites. Two XSs were surveyed in one river (Guden Å) in Denmark (DK), whilst three different rivers (Ätran, Kilan, Krokån) were surveyed in Sweden (SE). Different surveys were conducted with different instruments: UAS-borne OTT SVR100 ("UAS-SVR"), UAS-borne Particle Image Velocimetry ("UAS-PIV"), Static OTT SVR100 from bridge ("Static-SVR") and, as ground-truth, either the electromagnetic probe OTT MFpro ("MFpro") or the ADCP Teledyne StreamPro ("ADCP"). The river average depth, width, bed type and aquatic vegetation density are reported.

| Site Name | Location | Coordinates of Markers Lat, Long (°) | | Survey Date | Conducted Surveys | Average Depth (m) | River Width (m) | Bed Type and Aquatic Vegetation Density |
|---|---|---|---|---|---|---|---|---|
| | | Left Stream-bank | Right Stream-bank | | | | | |
| Guden_Svostrupvej | DK | 56.2233, 9.66901 | 56.22319, 9.669384 | 14 February 2020 | UAS-SVR UAS-PIV ADCP Static-SVR | 2.25 | 26 | Clayish, low vegetation density |
| Guden_GamleSkibelundvej | DK | 56.36587, 9.634325 | 56.36568, 9.63458 | 23 June 2020 | UAS-SVR UAS-PIV Mfpro | 1.96 | 24.3 | Clayish, high vegetation density |
| Ätran | SE | 57.27526, 12.99929 | 57.27526, 12.99856 | 28 May 2021 | UAS-SVR UAS-PIV | ≈1.8 | 44 | Rocky, very low density (or absence of) aquatic vegetation |
| Kilan | SE | 57.04974, 13.1166 | 57.04985, 13.11617 | 29 May 2021 | UAS-SVR UAS-PIV Mfpro | 0.63 | 28 | Rocky, very low density (or absence of) aquatic vegetation |
| Krokån | SE | 56.54607, 13.32273 | 56.54636, 13.32207 | 30 May 2021 | UAS-SVR UAS-PIV Mfpro | 0.54 | 52 | Rocky, very low density (or absence of) aquatic vegetation |

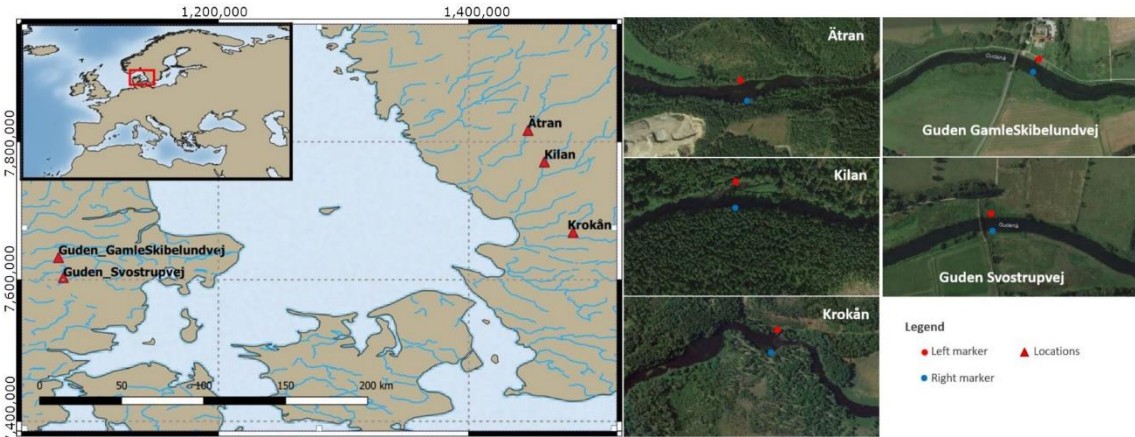

**Figure 2.** Map with the location of the different sites in Sweden and Denmark with detailed satellite images of the sites. Coordinate reference system is WGS84/Pseudo-Mercator. Source for the left-panel map is https://www.naturalearthdata.com/ (last access 3 January 2022), river outlines are from https://www.diva-gis.org/gdata (last access 3 January 2022), and the sites images are from https://www.google.com/earth/index.html (last access 3 January 2022).

### 2.3. Data Acquisition and Processing

#### 2.3.1. GNSS Processing

GNSS data were post-processed with the software RTKLIB Demo5 b34b from http://www.rtklib.com/ (last access 3 January 2022) [50] to obtain a Post-Processed Kinematic (PPK) solution (typically resulting in an accuracy of computed GNSS position of ca. 2–5 cm).

As basestation, a NovAtel Flexpack6 receiver with a NovAtel GPS-703-GGG pinwheel triple frequency (GPS and GLONASS) antenna was used.

### 2.3.2. PIV

The UAS platform was flown to the centre of the stream to retrieve a video of the flow. The post-processing workflow is similar to the one reported in [7]; we summarize as follows:

1. A sequence of frames of ca. 5–10 s was extracted from the video.
2. A MATLAB script was used to stabilize the video using natural stable features on the ground (e.g., rocks, land);
3. Lens distortion was corrected using the calibration-derived camera intrinsic parameters;
4. A segmentation method was applied to the video frames to convert the RGB frames into binary images, in which white is assigned to the seeding particles (natural or artificial), while black is assigned to the surroundings (e.g., water surface);
5. The surface velocity field was estimated with the image cross-correlation techniques implemented in the open-access software PIVlab for MATLAB [51]. A region of interest (ROI) was drawn where surface velocity across the XS had to be computed: the width of the ROI was equal to the river width and its length was ca. 4–5 m along the river course. In the PIV analysis, four different interrogation areas of size 256, 128, 64, and 32 pixels were chosen. After the PIV analysis, standard deviations (temporal standard deviation across video sequence) of the velocity vectors were analyzed: only a few frames showing clear outliers in velocity vectors (typically due to UAS vibrations or sudden movements) were discarded. A velocity field was extracted by computing the mean velocity vectors over the non-discarded frames.
6. The velocity in pixel units was then converted into metric unit. To this end, the coordinates of two markers (one pole on each streambank) were used to estimate the ratio between distance in pixels and distance in meters. The coordinates of two markers were found by identifying the position of the two markers in a previously acquired UAS-borne orthophoto.

Thus, compared to the previous methodology reported in [7], we applied an image segmentation method (simply based on identifying a thresholds, for the image RGB values, that can be used to separate seeding particles from the background). An example of the effect of image segmentation is shown in Figure 3. Figure 3 shows that the segmentation can separate between seeding and surrounding water, with a few misclassified areas (typically areas with sunglint, e.g., in the area under the bridge shown in the bottom panels of Figure 3). The image segmentation method more effectively separated seeding from surroundings than the standard image pre-processing algorithms used by PIVlab, such as histogram equalization, intensity high-pass filter and intensity capping.

Only in one river (Guden Å), the water surface did not contain a sufficient number of natural particles for successful PIV analysis, thus artificial seeding (woodchips) was thrown into the water by two operators standing on an upstream bridge. In all surveyed XSs in Sweden, the streams showed sufficient density of natural particles on the water surface (small floating wood particles, debris and especially foam generated by organic decomposition occurring in the nearby forests) and thus no artificial seeding was added.

### 2.3.3. SVR

At a short temporal scale (subsecond to few seconds level), river waves can be dominated by eddies and secondary currents. Thus if the sampling time of a Doppler radar was inadequate and eddies were large, the radar would measure the eddy-dominated velocities rather than the surface scatterers that are located on the surface of larger scale waves, which represent the river primary velocity [47]. Therefore, the SVR instrument has internal filters to smooth velocity measurements and cancel the effect of local eddies. Because of these internal filters, the SVR requires a long time (at least 20–30 s), during which the sensor is stable, to smooth velocity readings and accurately measure surface velocity. The SVR was

flown across the stream from the left to the right streambank, but shifted with respect to the XS line because the centre of the footprint of the SVR is not directly below the SVR (as a result of the SVR tilt). During the planned flight, the UAS was stopped at regular intervals (ca. every 1–2 m) to retrieve observations for ca. 1 min. The flight speed between those hovering intervals was as low as possible (ca. 0.5 m s$^{-1}$), and the flight direction was planned to keep the UAS nose pointing perpendicular to the flow; however, the uncertainty in flow direction and the inaccuracy in drone heading can generate an error of a few degrees in UAS orientation with respect to flow direction.

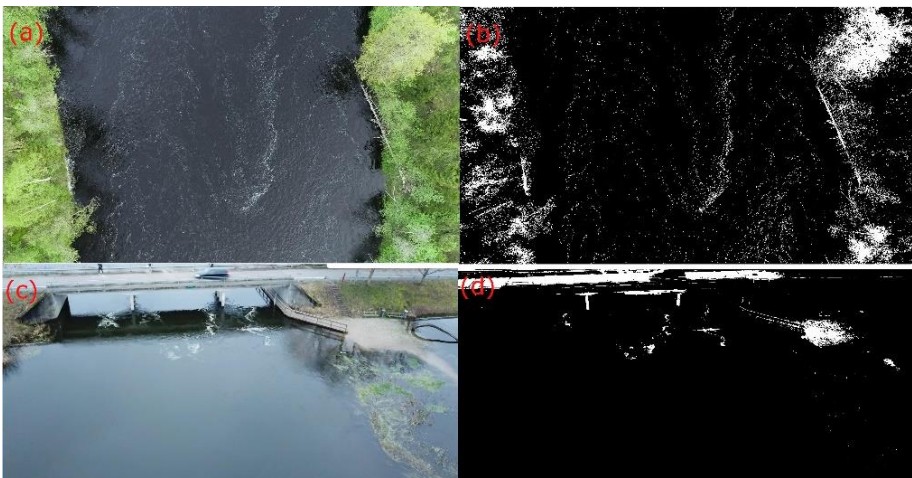

**Figure 3.** The image segmentation method was applied to the RGB video frames (left panel) to obtain a binary image (right panels): white color is assigned to seeding, whilst a black color is assigned to water surface (with a few outlier). (**a**,**b**) are in case of natural seeding (mainly foam), whilst (**c**,**d**) are in case of artificial seeding (woodchips).

SVR orientation (pointing upstream or downstream), sensor mounting angle, and flight height were parameters that required thorough testing. As sensor orientation, according to manufacturer recommendation and previous studies [31], upstream was preferred but downstream was also tested. As sensor mounting angle, values around 35° to 45° were tested because of the compromise between Doppler signal strength and footprint size. Flight height required careful evaluation for the following considerations:

1.  Although the sensor can theoretically measure from up to 20–25 m altitude above the water surface, the signal strength decreases with altitude. Thus, lower altitude is preferable in order to maintain good SNR.
2.  Lower altitude translates into a smaller footprint, which is a positive effect in terms of spatial resolution but a negative effect in terms of the size of the area in which measurements are averaged. In particular, smaller areas may also correspond to fewer scatterers, which may even result in a weaker signal.
3.  Furthermore, during flight at low altitude (less than 6–7 m above the water surface), the UAS-generated downwash induces significant turbulence on the water surface, thus affecting the water surface velocity (undesired effect) whilst at the same time increasing surface roughness (positive effect) [38].

Different tests were conducted to estimate the best flight height. From these tests, an optimal flight height of ca. 6 m above the water surface was identified because, at this flight altitude, the water surface velocity was not affected by the UAS downwash, similarly to [47].

Post-processing of the SVR data was performed using MATLAB scripts. Filtering of the observations was performed to include only the observations during which (i) SNR > 6 dBm, (ii) flow direction was consistent (depending on if the sensor was looking upstream or downstream), (iii) vibration index < 2, (iv) UAS was stable (less than 0.1 m s$^{-1}$ of UAS

velocity vector magnitude and less than 0.2 m from the target altitude) for at least 20 s, (v) vertical and horizontal GNSS-position standard deviation (an RTKLIB estimate about accuracy of each PPK-computed GNSS position) < 0.02 m.

2.3.4. Extraction of the Surface Velocity Profile along the XS

The velocity field had to be projected into the XS line (defined by the coordinates of the pole markers on the right and left streambanks). To this end, the XS was discretized in small intervals (25-cm wide). A similar method was then applied for PIV and for SVR:

- For PIV, each velocity vector retrieved at a maximum orthogonal distance to the XS line of 5 m (ROI length) was assigned to an interval of the XS line by nearest neighbor search. The magnitude of the velocity vector was computed. In case multiple vectors contained in the ROI length were assigned to the same interval, a median of the magnitude of those velocity vectors and a spatial standard deviation were computed. Thus, for PIV, the standard deviation is an estimate of the spatial variability, indeed it is computed among all the velocity vectors computed along the river course (inside the ROI) for each XS discretization interval.
- For SVR, velocity observations were filtered according to the maximum orthogonal distance from the XS line (only observations at a maximum distance of 5 m were included). A median (and standard deviation) was computed to average the SVR observations obtained during the sampling period during which the UAS-borne SVR was hovering in that XS interval. Thus, for SVR, standard deviation is computed from the velocity observations in the SVR sampling period: its computation is different compared to standard deviation for PIV.

## 3. Results

In this section, we show the results of the SVR and PIV and compare them with ground-truth observations.

### 3.1. Comparison Plots

Figure 4 shows the tests in Guden_Svostrupvej (DK). In this surveyed XS, the ADCP shows a mean velocity (at the measured level which is the closest to water surface, i.e., $\approx$0.2 m below surface) of ca. 0.5 m s$^{-1}$, with a median of ca. 0.56 m s$^{-1}$ and a maximum surface velocity of ca. 1 m s$^{-1}$. In this survey, PIV tends to underestimate the velocity (compared to the ADCP) in the middle of the XS, likely due to the following factors:

- The camera tilt angle was not nadir (tilted of ca. 30°) for complying with UAS legislation, which constrains UAS flights in the proximity of public roads (i.e., the bridge): the tilted camera could have introduced an uncorrected distortion effect affecting the pixel into metric unit conversion.
- Seeding was not uniformly distributed and tended to cluster.
- Wind (ca. 0.7 m s$^{-1}$) flowing against the river flow may have affected the seeding speed.
- ADCP does not measure velocity at the water surface but only $\approx$0.2 m below it. Given the average depth of this XS (ca. 2.25 m) and the low bed roughness (low vegetation density during the survey time and clayish bottom), wind is typically the main factors that may make surface velocity differ from velocity at 0.2 m depth.

The SVR seems to properly represent the velocity in the center part, but fails to measure velocity close to the edges. In particular, this occurs at the right edge, in which SVR-measured velocity rapidly increases to unrealistic values (ca. 2 m s$^{-1}$), while ADCP velocity shows values around 0.7 m s$^{-1}$. The values of PGA around 20 for the entire flight duration (and the detected signal power) could not explain the anomaly, partly because of these results, the SVR manufacturer decided to develop a new firmware version to output SNR level. However, the manufacturer explained that the anomaly on the right edge may be an effect of the roughness of the water surface that becomes too low (as visible also

on the video frames), causing unrealistic SVR velocities (too high). Indeed, according to manufacturer specifications, water roughness should be at least 1 mm (and preferably larger), and in case roughness was not sufficient, SVR velocity measurements could be unrealistic (either too high or too low).

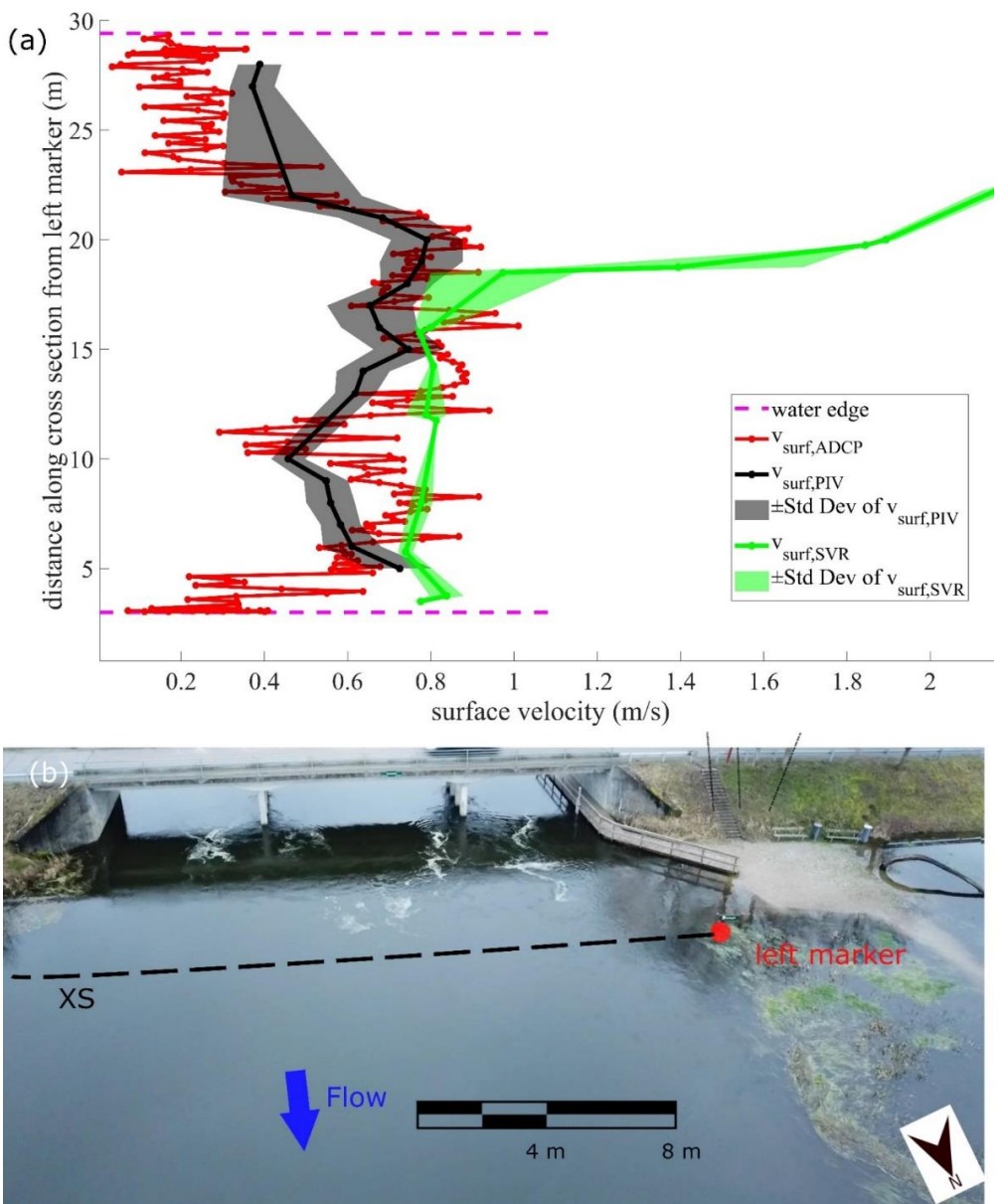

**Figure 4.** Guden_Svostrupvej XS (DK). (**a**) surface velocity plot comparing ADCP measurements of surface velocity ($v_{surf,ADCP}$), UAS-borne PIV ($v_{surf,PIV}$) and UAS-borne SVR ($v_{surf,SVR}$). SVR and PIV are represented with shaded grey (PIV) and shaded green (SVR) to present standard deviation ($\pm$Std Dev). (**b**) shows a UAS-borne video frame with also a red point indicating where the left marker was placed (North arrow, approximate flow direction and approximate scale are also shown).

Figure 5 shows the UAS-borne PIV, static and UAS-borne SVR (with the updated firmware enabling SNR output) tests conducted in Guden_GamleSkibelundvej (DK). In this surveyed XS, the MFpro shows a mean surface velocity of ca. 0.40 m s$^{-1}$, with a median of ca. 0.49 m s$^{-1}$ and a maximum surface velocity of ca. 0.60 m s$^{-1}$.

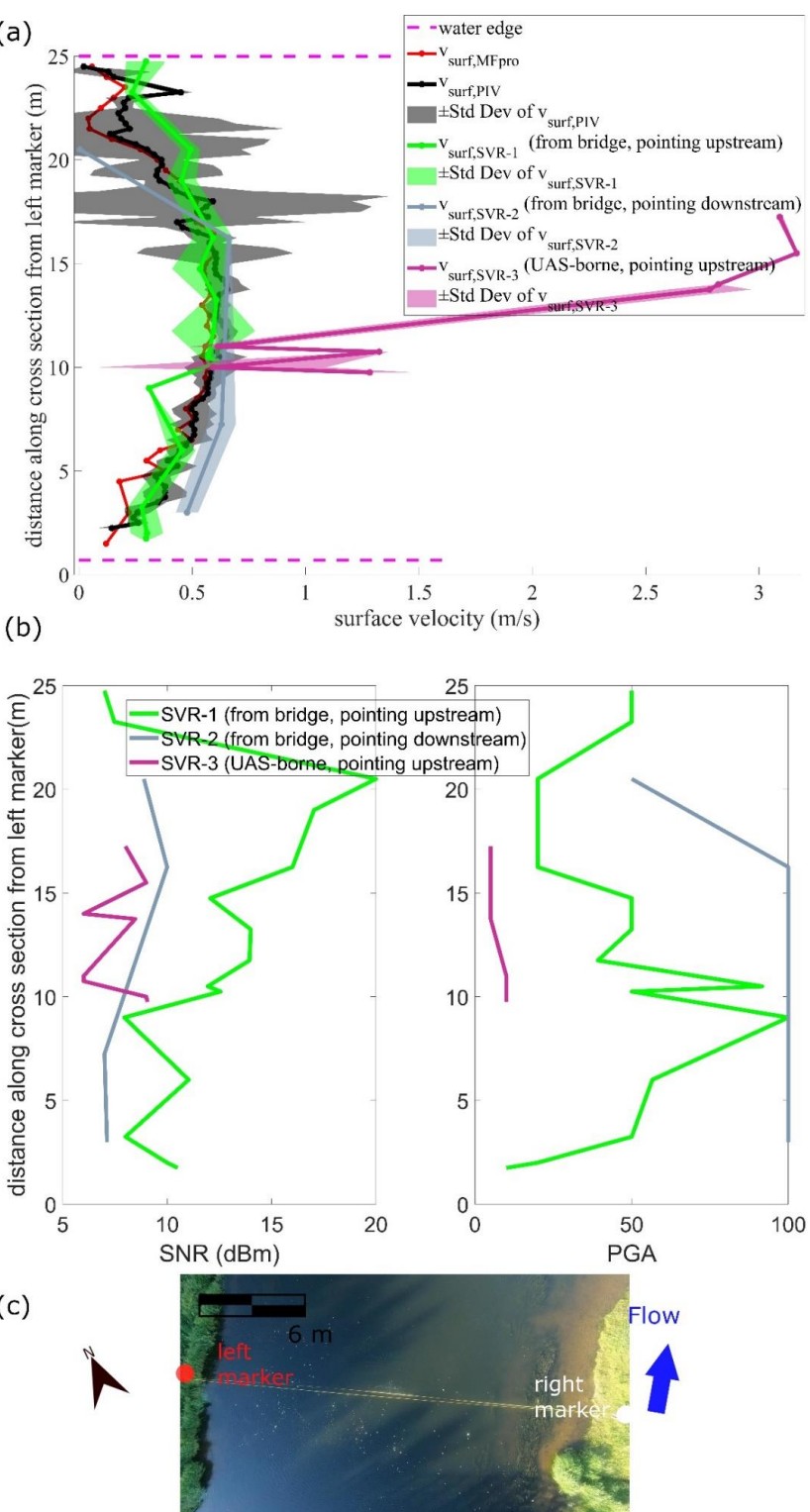

**Figure 5.** Guden_GamleSkibelundvej XS (DK). (**a**) Surface velocity comparing: MFpro measurements of surface velocity ($v_{surf,MFpro}$), UAS-borne PIV measurements ($v_{surf,PIV}$) and three SVR surveys. Two SVR are retrieved from bridge, i.e., $v_{surf,SVR-1}$ (pointing upstream), $v_{surf,SVR-2}$ (pointing downstream), whilst one from UAS, i.e., ($v_{surf,SVR-3}$, pointing upstream). SVR and PIV are represented with shaded regions to present standard deviation ($\pm$Std Dev). (**b**) Comparison of two quality indicators (SNR and PGA) for the three SVR surveys. (**c**) UAS-borne video frame of the XS (North arrow, approximate flow direction and approximate scale are also shown).

The PIV shows a good correlation compared to the surface velocity measured with MFpro. On the right streambank, the standard deviation (Std Dev) is rather high, which is caused by the high spatial variability along the river course, mainly caused by aquatic plants (also visible in the picture of Figure 5c). Static SVR tests were performed from the bridge by holding the SVR static; to obtain the velocity profile, the system was moved from one position to the other (at intervals of ca. 1 m) across the XS (with the system standing for 2 min at each position). Figure 5a shows the static test results for both the tests in which SVR pointed upstream and downstream. Because the upstream and downstream tests were conducted from the two sides of the bridge, the actual water surface area measured by the SVR footprint was different between the two tests, thus, surface velocity may slightly differ between the two cases. Both of these tests show a good agreement with MFpro. The test in which the SVR was pointed upstream shows better results (in terms of velocity comparison with ground-truth) than the downstream-looking test, especially in the center of the XS and on the left side. The advantage of pointing the SVR upstream is also shown by the SNR values (significantly higher in the upstream-looking test). The downstream-looking test observed a SNR < 7 on the right side of the XS (after ca. 20 m distance from left marker), thus those observations are omitted (as described in the Materials and Methods section regarding SVR filtering parameters). In contrast, the upstream-looking test shows a very high SNR value (ca. 20) at 20 m from the left marker and a sharp decline of SNR towards the right marker: the velocities detected in the upstream-looking test are inaccurate in the part of the XS after 20 m from left marker.

After these two static tests, additional static tests were performed a few hours later but all of them showed significantly lower SNR and unrealistically high velocity results (typically with velocity greater than 2 m s$^{-1}$): this may be an effect of decreased wind speed, which caused a reduced surface roughness. Tests with UAS-borne SVR also show very low SNR and unrealistic (too high) surface velocity measurements in most of the XS (apart from few points that gave realistic measurements). The UAS-borne test also shows a PGA value fluctuating between 5 and 10, which is lower than for the static test. A lower PGA may suggest that the SVR is disturbed by the UAS instability and high-frequency vibrations and, also because of low water surface roughness, is unable to detect surface water speed. Thus UAS-borne SVR tests in Denmark were not successful and additional tests were conducted in Sweden, in rivers with higher surface roughness.

Figure 6 shows the result for the Ätran (SE) river cross section. In this case, the river was fast flowing and too deep to wade across, thus no ground-truth was retrieved with MFpro and only UAS-borne PIV observations can be used as reference. The PIV shows a low velocity portion in the middle of the XS, where a submerged bar is located and flow is thus significantly reduced. In this XS, three UAS-borne SVRs tests were conducted. UAS-borne SVR shows significantly higher SNR than in the UAS-borne tests in Denmark, as a consequence of the high surface roughness. The UAS-borne tests also show a very low PGA, which indicates a very strong signal, but PGA below 5 may also indicate SVR saturation and thus reduced accuracy. The first SVR test shows relatively good correspondence with PIV in the high velocity portion but poor matching with PIV in the area where the submerged bar is. The second and third SVR surveys show erratic results, with a large mismatch compared to the first survey. Surprisingly, both the second and third SVR surveys show a low-velocity measurement around 11–12 m from the left marker, in complete disagreement with PIV and the first SVR survey.

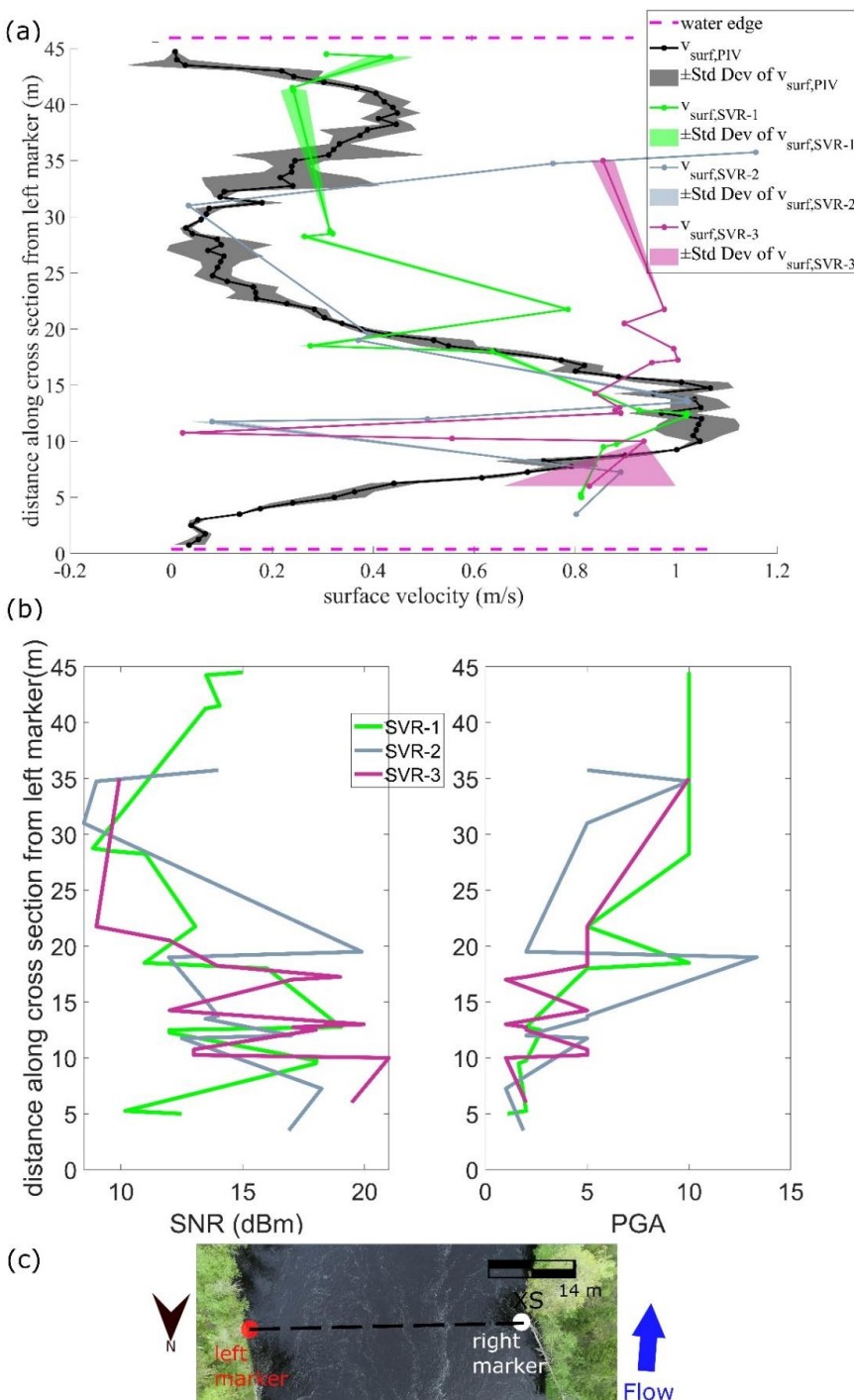

**Figure 6.** Ätran XS (SE). (**a**) surface velocity comparing: UAS-borne PIV measurements ($v_{\text{surf,PIV}}$) and three UAS-borne SVR ($v_{\text{surf,SVR-1}}$, $v_{\text{surf,SVR-2}}$ and $v_{\text{surf,SVR-3}}$, all of which pointing upstream) surveys. SVR and PIV are represented with shaded regions to present standard deviation ($\pm$Std Dev). (**b**) Comparison of two quality indicators (SNR and PGA) for the SVR surveys. (**c**) UAS-borne video frame of the XS (North arrow, approximate flow direction and approximate scale are also shown).

Figure 7 shows the test in Kilan (SE). In this river, a small non-submerged bar divides the stream into two braided channels. In this surveyed XS, the MFpro shows a mean surface velocity of ca. 0.38 m s$^{-1}$ and a median of ca. 0.35 m s$^{-1}$. Velocities significantly differ between the two braided channels: the maximum surface velocity is 0.94 m s$^{-1}$ in the left channel, whilst the right channel has a maximum surface velocity of 0.44 m s$^{-1}$. In this case, PIV shows a good agreement with the acquired MFpro throughout the XS. The first SVR

flight shows good agreement with MFpro in the channel with high velocity (apart from one SVR outlier at around 8 m from the left marker). The first SVR flight shows faulty readings when part of the SVR footprint lies on the bar; indeed, soil or grass in the SVR footprint can introduce large error sources (e.g., grass fluctuation caused by wind). Furthermore, the first flight is also unable to retrieve observations in the right side of the river (because of lower surface roughness resulting in SNR < 7, thus observations were omitted). The second SVR flight shows significantly lower surface velocity (ca. 15–20% less) than the first SVR flight in the left side of the river (higher flow portion) and also shows faulty readings in the portion where the bar lies and on the right side of the river. Thus, the repeatability of UAS-borne SVR results is questionable.

The repeatability of UAS-borne SVR results is also poor in the XS in Krokån (SE) river, shown in Figure 8. In this surveyed XS, the MFpro shows a mean surface velocity of ca. 0.29 m s$^{-1}$, with a median of ca. 0.17 m s$^{-1}$ and a maximum surface velocity of ca. 1.07 m s$^{-1}$. Three SVR flights were conducted on this site. During the first flight, the SVR was pointed downstream and the SNR was the poorest among the three flights. The second flight (pointing upstream) gave the best results in terms of velocity comparison with MFpro, but velocity in the slow flow portions is significantly overestimated with SVR. The third flight (pointing upstream) matches poorly with the second flight, even though SNR and PGA indicators are similar. Moreover, in this XS, UAS-borne PIV matched well with MFpro; however, PIV and MFpro surface velocity measurements differ close to the right bank (at ca. 45 m from left marker), and this can be caused due to the lack of seeding in that portion but especially due to flow heterogeneities along stream direction, which are caused by the small bar close the streambank.

### 3.2. Statistics

The UAS-borne SVR showed poor repeatability between the surveys at the same XS and very large errors at low velocity, with land or vegetation in the SVR footprint (e.g., close to the streambanks, bars in the middle of the streams) or in the locations with low surface roughness, often resulting in unrealistic velocity measurements. Because of these problems, a statistical comparison of SVR velocimetry results and results from other methods are not meaningful. On the other hand, UAS-borne PIV showed good agreement with MFpro, thus statistics were computed.

Table 2 shows the Mean Absolute Error (MAE), Mean Bias Error (MBE) and Root Mean Square Error (RMSE) computed by comparing the surface velocity observations retrieved by PIV (interpolated at MFpro spatial resolution) with the observations retrieved with MFpro. Table 2 shows errors in the order of a few cm s$^{-1}$, apart from the first XS in Guden_Svostrupvej (DK) in which a MAE of ca. 14 cm s$^{-1}$ is computed. The R$^2$ coefficient is greater than 0.85 in two XSs, ca. 0.70 in Krokån and ca. 0.67 in Guden_Svostrupvej (DK).

**Table 2.** Comparing the surface velocity (v$_{surf}$) measurements retrieved with PIV and ground-truth (MFpro for all XS, apart from Guden_Svostrupvej where ADCP was used). The ground-truth maximum, minimum and mean of v$_{surf}$ values are reported as reference. In Ätran ground-truth measurements were not acquired, thus statistics are not shown. The following statistics comparing PIV and ground-truth measurements are shown: R$^2$ (coefficient of determination), MAE (Mean Absolute Error), MBE (Mean Bias Error), RMSE (Root Mean Square Error).

| Site Name | Max v$_{surf}$ (m s$^{-1}$) | Min v$_{surf}$ (m s$^{-1}$) | Mean v$_{surf}$ (m s$^{-1}$) | R$^2$ - | MAE (m s$^{-1}$) | MBE (m s$^{-1}$) | RMSE (m s$^{-1}$) |
|---|---|---|---|---|---|---|---|
| Guden_Svostrupvej | 1.01 | 0.01 | 0.50 | 0.67 | 0.14 | −0.01 | 0.16 |
| Guden_GamleSkibelundvej | 0.60 | 0.04 | 0.40 | 0.88 | 0.06 | 0.04 | 0.07 |
| Ätran | - | - | - | - | - | - | - |
| Kilan | 0.94 | 0.00 | 0.38 | 0.86 | 0.08 | −0.02 | 0.11 |
| Krokån | 1.07 | −0.02 | 0.29 | 0.70 | 0.11 | −0.01 | 0.15 |

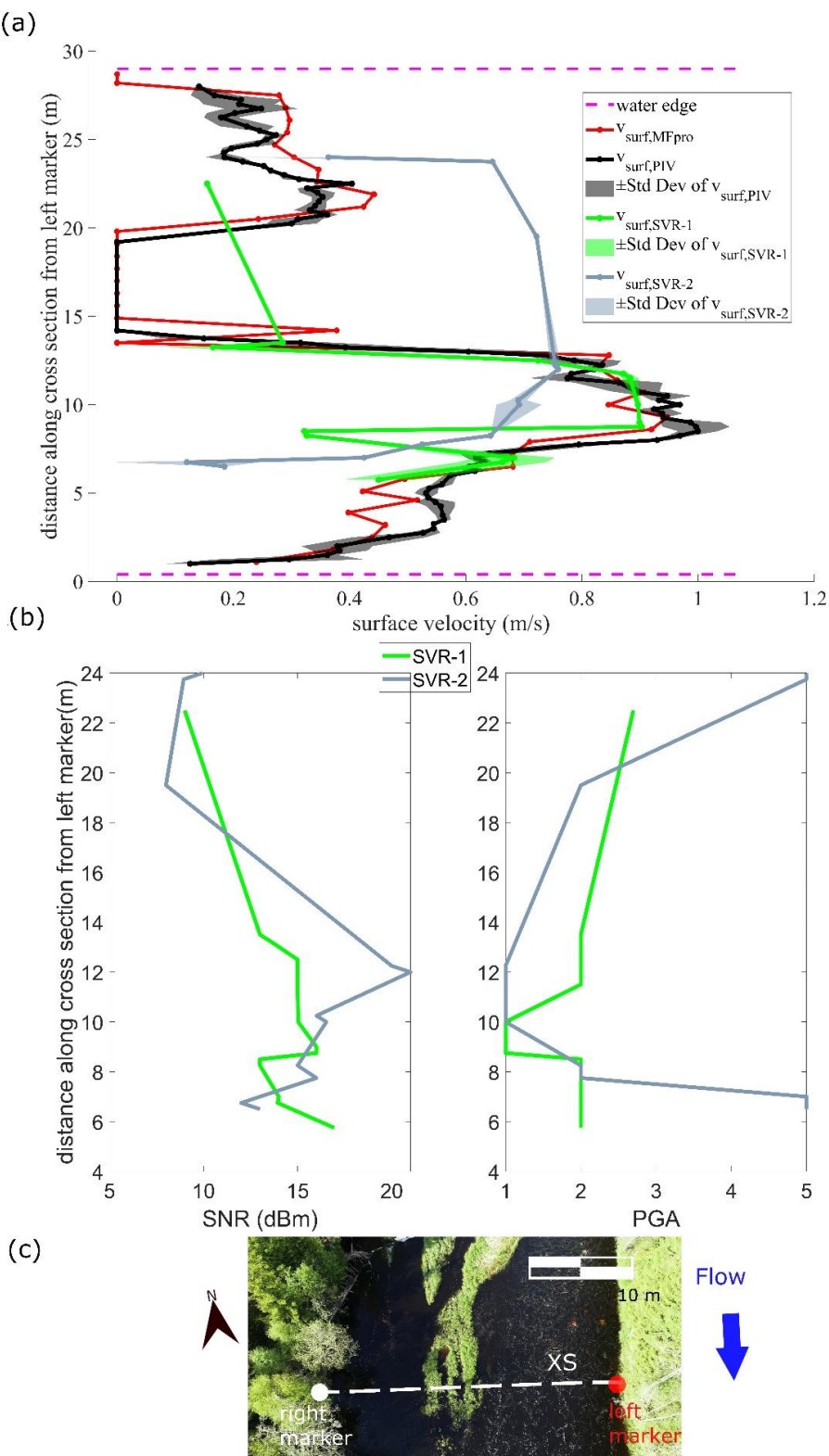

**Figure 7.** Kilan XS. (**a**) surface velocity comparing: MFpro measurements of surface velocity ($v_{surf,MFpro}$), UAS-borne PIV measurements ($v_{surf,PIV}$) and two UAS-borne SVR ($v_{surf,SVR-1}$, $v_{surf,SVR-2}$, all of which pointing upstream) surveys. SVR and PIV are represented with shaded regions to present standard deviation (±Std Dev). (**b**) Comparison of two quality indicators (SNR and PGA) for the SVR surveys. (**c**) UAS-borne video frame of the XS (North arrow, approximate flow direction and approximate scale are also shown).

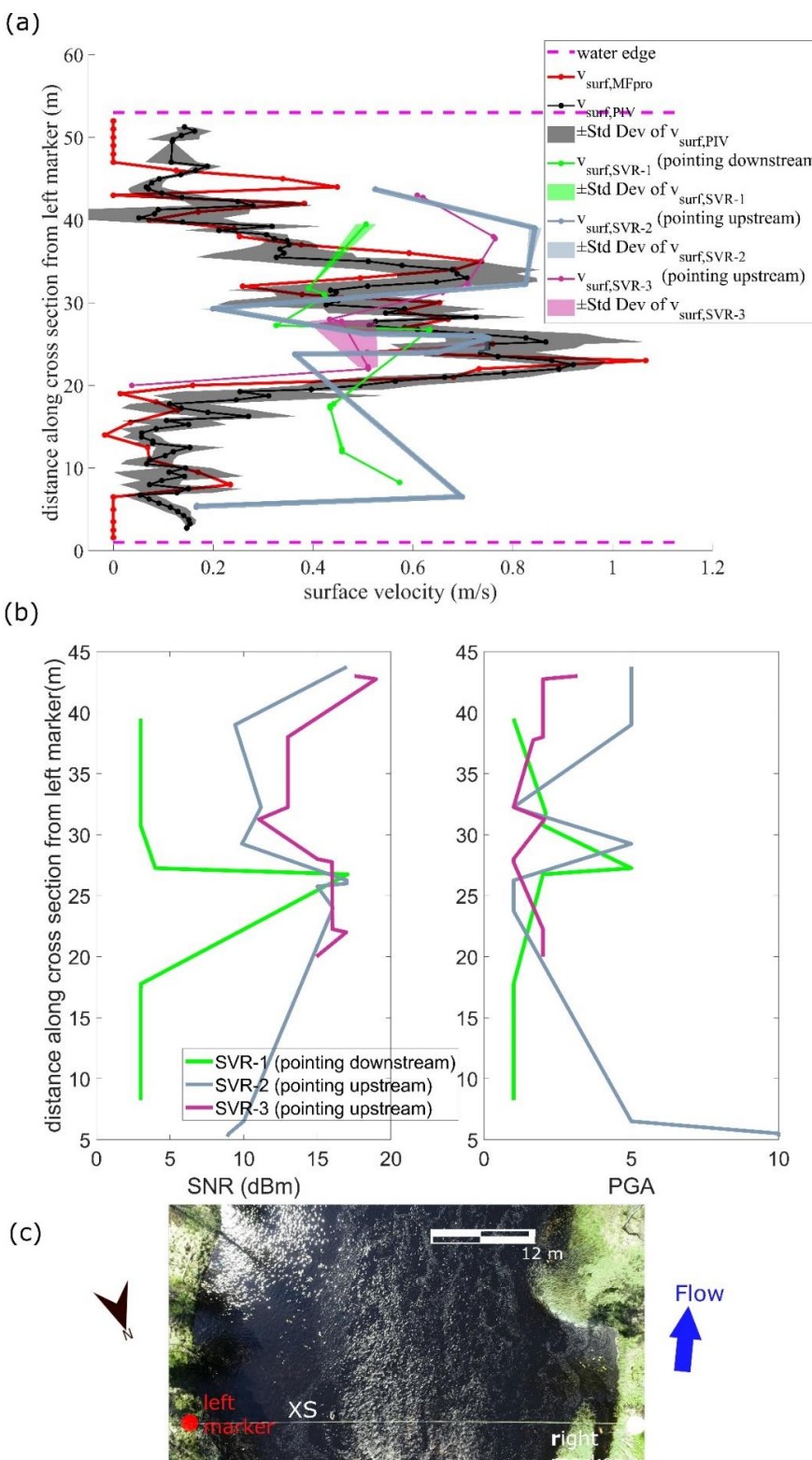

**Figure 8.** Krokån XS (SE). (**a**) surface velocity comparing: MFpro measurements of surface velocity ($v_{surf,MFpro}$), UAS-borne PIV measurements ($v_{surf,PIV}$) and three UAS-borne SVR ($v_{surf,SVR-1}$ pointing downstream, $v_{surf,SVR-2}$ and $v_{surf,SVR-3}$ pointing upstream) surveys. (**b**) Comparison of two quality indicators (SNR and PGA) for the SVR surveys. SVR and PIV are represented with shaded regions to present standard deviation (±Std Dev). (**c**) UAS-borne video frame of the XS (North arrow, approximate flow direction and approximate scale are also shown).

## 4. Discussion

We discuss the results of the two techniques (PIV and SVR) and the advantages and disadvantages of both.

### 4.1. PIV

#### 4.1.1. PIV Advantages

PIV has several advantages compared to SVR, including the following:

1.  PIV estimates both magnitude and angle of the velocity vector (thus can detect velocity along and across the XS).
2.  PIV estimates the velocity field over the entire XS width in one single video and also gives information about the spatial variability along the portion of the river course that is visible in the video.
3.  PIV estimates velocity with a high spatial resolution (typically dependent on the interrogation area size).
4.  PIV videos can be stabilized so that the effects of UAS vibrations and drifts are reduced.

#### 4.1.2. PIV Limitations

PIV has also some significant limitations:

1.  Seeding characteristics, such as density and distribution, control the quality of PIV estimates [21,24]. Artificial seeding was used in the Danish streams, in which the number of trackable natural features on the water surface was insufficient. Thus, in those conditions, PIV required one operator for flying the UAS and multiple operators to add artificial seeding. This was not the case in Swedish streams, in which a sufficient number of natural particles or foam was present. However, if the streams were surveyed during different seasons, the foam (generated by decomposition processes) may be not visible, thus artificial seeding may be required.
2.  PIV required a longer processing time than SVR, velocity probe, or ADCP measurements— ca. 20–30 min per survey when PIV is applied. Indeed, even though most post-processing can be automatized, time is still required for video editing, sequence selection, stabilization and PIVlab processing.
3.  PIV is cumbersome in very large rivers: the surveyed rivers were only a few tens of meters wide (the widest XS was ca. 52 m); however, if the rivers were several tens or hundreds of meters, it would be impossible to cover the entire river width with a single video and thus multiple videos should be acquired, causing additional complexity in (i) georeferencing every single video or/and mosaicking multiple videos, (ii) stabilizing videos in which stable features on streambanks are not visible, and (iii) adding artificial uniform seeding over large river widths in cases where natural seeding is insufficient.
4.  The velocity at which seeding moves is directly affected by wind. This is especially critical for lightweight tracers, which are more affected by wind because a large portion of their volume is above the water surface.
5.  PIV using optical cameras only works under good daylight conditions and fails during the night.

#### 4.1.3. PIV Results

PIV velocimetry results, when compared to MFpro, show small errors, typically only few cm s$^{-1}$ and R$^2$ typically greater than 0.88, apart from two XS (Guden_Svostrupvej and Krokån). In Guden_Svostrupvej (R$^2$ of 0.67, MAE of ca. 14 cm s$^{-1}$ and RMSE of 16 cm s$^{-1}$), the surface velocity is underestimated with PIV in the center of the XS and overestimated on the right side of the XS, mainly due to camera tilt angle (distortion effect), poor seeding density, wind effects and perhaps also because ground-truth was acquired with ADCP, which has a blind zone (unmeasured zone close to water surface) larger than MFpro. In Krokån (R$^2$ of 0.70, MAE of 11 cm s$^{-1}$ and RMSE of 15 cm s$^{-1}$), the lower R$^2$ and the higher

RMSE is caused by lack of seeding and flow heterogeneities in the portion close to the right bank caused by the small sediment bar located in that area, whilst the measurements in the center and close to the left bank are accurate.

4.1.4. Comparison of PIV Results with Previous Studies

The estimated PIV accuracy is in agreement with previous studies. For example, Dal Sasso et al. [52] found RMSE of 0.1–0.2 m s$^{-1}$ (with large percentage error at low velocities) when comparing UAS-borne PIV estimates and velocity probe estimates. Strelnikova et al. [6] also found errors in UAS-borne PIV compared to ground-truth in the order of few cm s$^{-1}$ or dm s$^{-1}$, with significantly higher percentage errors for low velocities.

De Schoutheete et al. [49] found that PIV significantly overestimates velocity compared to MFpro estimates (because MFpro has to be fully submerged in order to estimate velocity), with an average bias of ca. 0.86 that was introduced as a constant bias correction by the authors. Such a large bias was not found in our case, but in three out of four tests, PIV slightly underestimated velocity compared to MFpro.

The image segmentation method applied to the video frames, which was used to separate the seeding from the surroundings, allowed focusing on portions where seeding was present. The developed segmentation method was especially useful in the XSs where no artificial seeding was added or with non-uniform seeding density because it avoids averaging real measurements with false zero velocities and consequently underestimating velocity, as reported in [3,6,24].

*4.2. SVR*

SVR has some important advantages compared to PIV, but also some major disadvantages.

4.2.1. SVR Advantages

The main advantages of SVR compared to PIV are as follows:

1. SVR does not require seeding on the water surface.
2. SVR processing time is deterministic and can be fully automatized, making it suitable also for real-time velocity determination (e.g., during floods).
3. SVR is ideal to be applied in large rivers (e.g., several tens or hundreds of meters wide), especially when the SVR footprint is small compared to the river width. In this case, a flight conducted at 10–15 m altitude across an XS could measure the entire horizontal velocity profile.
4. SVR performance is independent of daylight conditions.

4.2.2. SVR Limitations

SVR also showed several limitations:

1. The SVR (and Doppler radar instruments in general) requires water roughness for receiving a sufficient amplitude of backscattered signal. This may be critical over lowland streams which have soft riverbed materials, mild slopes and velocities. In those rivers, the main source of surface roughness would be wind.
2. Microwave antennas of Doppler radar instruments have a beam-width on the order of a few tens of degrees. The SVR footprint is large (12° Azimuth, 24° Elevation): only one averaged observation is output for the entire footprint. This leads to a poor spatial resolution and becomes critical when the velocity changes rapidly across the river width.
3. The SVR only measures the velocity of waves traveling toward or away from the antenna in the plane of incidence, so it cannot determine the velocity in all the other directions.
4. The tested SVR instrument has internal filters that are optimized for static deployment. Despite the choice of filter type (moving average or infinite impulse response) and filtering window, our tests show that the SVR needs a long time (at least 20–30 s) during which the SVR is stable before the velocity measurements are smoothed, as

also reported by Fulton et al. [47]. This can become critical for UAS-deployment, in which drone vibrations and drifts have a direct impact on measured velocity.

5. The SVR velocity measurements could not be corrected for drone vibrations and drifts. Because the UAS-borne Inertial Measurement Unit and GNSS system can measure UAS velocity, roll angle, pitch angle and heading, ideally the Doppler radar measurements should be corrected for these known drone movements. However, this would require the Doppler radar to output raw signal data, instead of processed velocity (as for the deployed SVR).

### 4.2.3. SVR Results

The observations acquired with SVR static from a bridge (only tested in Guden_ GamleSkibelundvej XS) showed a good match to the MFpro (especially for the case of SVR pointing upstream). However, the water roughness in that specific XS was at the minimum limit; indeed, static tests that were performed a couple of hours later, when wind speed was lower (according to wind measurements of meteorological stations located nearby the survey area), resulted in too low SNR and unrealistic measurements.

The observations acquired with UAS-borne SVR in Danish rivers showed unrealistic velocity, apart from a portion of the test in Guden_Svostrupvej (DK) in which velocity measurements show a proper match with MFpro. The unrealistically high (2 m s$^{-1}$) UAS-borne SVR measurements are not directly caused by the UAS velocity; indeed, measurements are acquired by flying the UAS to the hovering position at low speed (ca. 0.5 m s$^{-1}$) and then make it hover for typically 1 min. The first 20 s of hovering were discarded from the SVR observations; indeed, the SVR sensors need to be static for least 20 s to infer the optimal parameters (e.g., PGA) and stabilize its measurements. However, the UAS vibration and drift can affect SVR internal processing, including the PGA settings, especially when the water roughness (and thus the SNR) is marginal. The observations retrieved in Swedish rivers with UAS-borne SVR show significantly higher SNR and lower PGA (especially in the high velocity portions). However, despite a few SVR flights showing a good match with MFpro in the high flow portions, the repeatability of SVR measurements retrieved during different SVR flights is poor. In general, we suggest that the main error sources are related to the following:

1. The uncertainty in UAS navigation (typically a couple of meters) and in UAS height (ca. 0.5 m), together with the large SVR scanning angle, that may cause that the water surface areas illuminated by the SVR antennas are different.

2. The uncertainty in UAS heading angle (ca. 5°) that may cause misalignment between the UAS-borne SVR and the flow direction of the river.

3. Disturbance caused by the SVR antenna beam illuminating portions of soil or vegetation, as also reported in Fulton et al. [47]. This was the case for the test conducted in the Kilan River (SE), which showed that the bar in the center of the river caused unrealistic velocity readings, and in all the measurements conducted close to the streambanks. Thus, larger river widths without rapid changes of water velocity across the XS should be preferred in site selection because they allow for having semi-constant surface velocity in the SVR footprint.

4. Errors caused by multipath, e.g., radiation hitting the UAS and further reflected towards water, should be further examined because they can be substantial as the UAS is made of conductive carbon fiber.

5. The sites in Denmark showed too low water roughness (low SNR). However, in the Swedish tests, PGA is typically less than 5, which, according to the manufacturer, may mean that the reflected signal is very strong, and the receiver may be oversaturated, which could result in reduced accuracy. Thus, sites should be selected carefully because high-water roughness (e.g., strong macro-turbulence) can limit the accuracy, whilst too low water roughness results in unreliable measurements. Ideally, according to Melcher et al. [38], water roughness should be preferably caused by wind.

### 4.2.4. Wind Effect and Flight Altitude

High wind speed has a positive effect on water roughness; however, it affects drone stability, which may cause large inaccuracies in UAS-borne SVR readings, and in addition, wind affects water surface velocity. The magnitude of wind effect on surface velocity measurements performed by a Doppler radar was evaluated by Plant et al. [31], taking into account the effective depth at which water velocity is measured by microwave measurements. According to Plant et al. [31], the magnitude of the wind drift in the wind direction at the surface is ca. 2% of the wind speed measured at 10 m altitude. For a $10 \text{ m s}^{-1}$ wind speed, this corresponds to $20 \text{ cm s}^{-1}$, which is not negligible. When there was no wind to increase the surface roughness, Melcher et al. [38] flew the manned helicopter at a lower altitude and used the propeller-generated downwash to increase the water surface roughness. However, in that research, the authors could deploy two antennas, one on each side of the helicopter, to remove the water velocity components caused by the propeller downwash from the actual flow velocity (indeed downwash-generated waves always move away from the helicopter). Deployment of two antennas on both sides of the UAS was not possible here, thus the water velocity effect caused by low UAS flights could have directly affected SVR readings. However, the downwash generated by a UAS is significant but does not have the same intensity and distance of influence as for a manned helicopter. A few tests (not shown in this paper) were performed in Guden_GamleSkibelundvej (DK) XS at altitude as low as 1.5 m above the water surface; however, the SNR reported during those UAS-borne flights was lower than 6–7 dBm and the velocity measurements were also unrealistically high (greater than $2 \text{ m s}^{-1}$).

### 4.2.5. Comparison of SVR Results with Previous Studies

Comparison can be made with helicopter deployment [38] and UAS-borne deployment [47]. Melcher et al. [38] found good agreement between helicopter-borne Doppler radar and ground-truth, typically in the order of a few decimeters $\text{s}^{-1}$. However, the measured velocities were significantly higher (up to $2.5 \text{ m s}^{-1}$), and the river was wider (ca. 100 m) than at our sites, thus we may expect that faster flow and the wider river geometry conditions were more favorable. Fulton et al. [47] found an excellent correlation between velocity measured with a UAS-borne Doppler radar and velocity measured with a handheld velocity radar as well as an acoustic Doppler velocimeter; however, only the maximum cross-section surface velocity was reported with UAS-borne SVR and the surface roughness of the sites was unknown.

### 4.3. Future Research

For PIV, we suggest that future research should be conducted to make the seeding process more autonomous, for example by using a secondary UAS that releases seeding in a direction at a specific angle with respect to flow in order to obtain rather uniform seeding density when seeding flows through the investigated XS.

For SVR, more tests should be conducted in streams with different surface roughness conditions, especially in situations in which surface roughness is mainly caused by wind.

Tests should be conducted in rivers with different hydrological and morphological conditions (e.g., large rivers with low gradient of surface water velocity across the river width). Furthermore, UAS choice is critical because the platform has to be very stable also in windy conditions. Additionally, flight missions should be performed in fully automatic flight mode with RTK GNSS to ensure centimetric accuracy in navigation and at least 1–2° heading accuracy. The error caused by multi-path (e.g., from UAS structure) should be further evaluated.

Moreover, it would be crucial to test a Doppler radar able to output IQ (In-phase and quadrature components) samples or at least frequency, amplitude (SNR) and phase of the received signal, preferably at high temporal resolution. Frequency information would be essential for customizing post-processing workflows for UAS deployment (e.g., correct for drone vibrations). Frequency, together with SNR, is necessary for spectrogram visualization,

specifically to evaluate the signal quality of the detected velocity. Phase signal information from the different radar receiving antennas would be essential to estimate range and especially angle of reflected signal in order to identify the location inside the radar footprint from which the signal is received and to distinguish between water and surroundings.

## 5. Conclusions

A commercial 24 GHz Doppler radar, developed for static deployment, was readapted for UAS-borne use for estimating the horizontal flow velocity profile across river cross-sections. Furthermore, the velocity profile was also estimated with UAS-borne PIV.

Different tests were conducted in two Danish XSs (Guden Å stream), characterized by mild slopes, soft riverbed material and low water roughness, and in three Swedish rivers, characterized by relatively steep slopes, hard riverbed material and high-water roughness. We compared PIV and SVR measurements and benchmarked them against ground-truth acquired with an electromagnetic velocity probe (MFpro) or with ADCP:

- PIV matched well with ground-truth in the Swedish rivers (where natural seeding was sufficient) and in Denmark (where artificial seeding was added) with typical velocity errors of 10 cm s$^{-1}$ or better and R$^2$ typically larger than 0.7 (specifically larger than 0.85 in two XSs). Only in one XS were the errors larger (RMSE greater than 15 cm s$^{-1}$ and R$^2$ equal to 0.67) because of wind, clustering seeding, non-nadir looking camera (and potentially because in that case, ground-truth was acquired with ADCP, which can measure velocity only ca. 0.2 m below the surface). The static SVR test (from bridge) conducted in Guden Å showed good agreement (differences typically smaller than 10 cm s$^{-1}$) with ground-truth velocity measurements (apart from the portion close to the streambanks).
- The UAS-borne SVR tests conducted in Guden Å showed unrealistic velocity observations, apart from a portion of the one XS (Guden_Svostrupvej), which showed good agreement with ground-truth.
- The UAS-borne SVR conducted in Sweden showed high SNR and, in some cases, good agreement with ground-truth in the faster flowing portions. However, different SVR flights showed significant differences in SVR measurements, thus the repeatability of the results should be further investigated.

Our research suggests that, in order to customize UAS deployment of Doppler radar, a radar providing raw output (frequency, signal amplitude or SNR and phase of all radar receivers) would be needed, instead of a Doppler radar such as the deployed SVR, which is developed for static use and outputs processed data of water surface velocity.

**Author Contributions:** Conceptualization, F.B. and P.B.-G.; methodology, F.B., M.C.F., J.L., K.S., S.K. and P.B.-G.; software, F.B., K.S. and S.K.; validation, F.B. and P.B.-G.; writing—original draft preparation, F.B.; writing—review and editing, F.B., M.C.F., J.L., K.S., S.K. and P.B.-G.; funding acquisition, F.B. and P.B.-G. All authors have read and agreed to the published version of the manuscript.

**Funding:** This work was supported by the projects RIVERSCAPES (file number 7048-00001B) and CHINAWATERSENSE (file number 8087-00002B) funded by the Innovation Fund Denmark (Innovationsfonden) and by the National Key Research and Development Program of China project (2018YFE0106500).

**Data Availability Statement:** The acquired data are available online in the repository archived in Zenodo.org (https://doi.org/10.5281/zenodo.5149856). The repository contains the videos used for the PIV velocimetry, the PIV results, the processed SVR data and the MFpro measurements.

**Acknowledgments:** We thank Stefan Siedschlag from OTT HydroMet (Germany) for lending the OTT SVR100 and for all the provided instrument support.

**Conflicts of Interest:** The funders had no role in the design of the study; in the collection, analyses, or interpretation of data; in the writing of the manuscript, or in the decision to publish the results.

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
