# Peer review of "Challenges with Regard to Unmanned Aerial Systems (UASs) Measurement of River Surface Velocity Using Doppler Radar"

_remotesensing, doi:10.3390/rs14051277_

Round 1
Reviewer 1 Report
Dear Authors,
Thanks for your clarifications regarding to general remarks, also accepting minor comments. I’m still think that in the manuscript more emphasis should be placed on explaining the local morphometric conditions of the river channel to the presented results.
You added a many supplementary information to the manuscript, especially regarding the methodology, which significantly contributed to the understanding of the measurements.
I suggest work on figures quality, before eventual publication, should be developed qualitatively, e.g. smaller font on Figure 1 and 2, resize figure 2, add source of basemap on figure 2, etc.
Best regards
Author Response
Dear editor,
enclose the re-submission of paper "Challenges with regard to Unmanned Aerial Systems (UASs) measurement of river surface velocity using Doppler radar".
We would like to thank the editor and all the reviewers for their comments and valuable suggestions. We have incorporated the requests of the reviewers. The detailed reply to the reviewers can be found in the dedicated section.
Best regards
Reviewer 2 Report
The paper discusses the reliability of Unmanned Aerial Systems (UAS)-borne Doppler radar and UAS-borne Particle Image Ve-14 locimetry (PIV) in different rivers for the measurement of river water velocity. Although the UAS-borne Doppler radar platform could get fast and low-cost surveys in remote locations, however, the reliability of the measurements of the 24 GHz Doppler radar are relatively low. PIV could reconstruct the velocity profile with high accuracy in all investigated rivers. The experimental results are sufficient and sound. I recommended the paper for publishing.
Author Response
Thanks for your time in reviewing the manuscript.
Best Regards.
Reviewer 3 Report
The paper tested a doppler radar aboard a UAS, to measure water surface velocity in several Danish rivers.
I found the paper difficult to read in form even though the analysis is quite clear. Therefore, the language of the text should be revised because it is redundant and often unclear.
The focus of the article defined in the introduction (which is also highlighted in red as evidence of the previous review), suggests a study that shows and demonstrates the importance of having a radar that provides frequency and amplitude along with velocity. However, the study does not show this, but highlights the limitations of the selected technology. I would fix this in the abstract, introduction, and conclusion.
Author Response
Thanks for your time in reviewing the manuscript. Regarding your comments:
We have significantly improved the manuscript language and also the quality of all figures. We agree that the study does not directly show the advantage of having raw Doppler signal frequency, amplitude and phase measurements. For this reason, we remove that from the abstract and introduction. However, we think it is important to remark that in the discussions and also briefly in the conclusions as we want to highlight that it is our suggested research path to obtain more reliable measurements.
Best Regards.
Reviewer 4 Report
Dear Authors,
your study is interesting and in line with the scope of the Journal, but I suggest revising it further to better clarify some points and provide more clear outcomes.
Apart from the following general comments, please refer to the attached pdf for further hints.
In general, the text is rather vague and more details are needed, especially to guarantee the study reproducibility and allow readers to properly judge your work.
In the Results section, I suggest firstly introducing the acquired data, instead of starting by comparing them. How can a reader catch what is the magnitude of the measured data or what are the "right" ones? This also reflects in subsection 3.2: how can a reader see that SVR velocimetry results are not reliable?
I suggest improving the Discussion section, reviewing studies made on similar rivers (if any) to better evaluate the pros and cons of the proposed methods.
Please provide better figures, as the quality of the present ones is rather poor.

Author Response
Thanks for your time in reviewing the manuscript. Regarding your comments:
- The pdf with detailed replies to your specific comments was attached here as submitted material: the paper was modified accordingly to those suggestions.
- The manuscript language has been significantly improved.
- The quality of all figures was improved; in particular, the figure with the location map was significantly changed; furthermore, the bottom panels of the figures showing the acquired measurements now also show the flow direction, XS line and a spatial scale.
- In the Results section, we now introduce the ground-truth acquired data (e.g. maximum, mean, median surface velocity) before comparing them with SVR or velocity probe for each site. We also modified the Table 2 to include the R2 statistics and the mean, max and min surface velocity for each site.
- We think that we already cite studies conducted with PIV and SVR with rivers of similar sizes in the introduction and in the discussion parts. However, we did not find published studies referring to surface velocity measurements in the river we surveyed or nearby rivers.
Best Regards.

Reviewer 5 Report
Authors have provided quite interesting scientific, but also practical research by using the drones with radar measuring devices for the purpose of measuring the surface velocity profiles, although that surface profiling is not exact from the engineering point of the hydrometry. Paper is well structured, idea and methodology justified and explained very well. I am proposing a minor revision.
-authors explained that they have increased turbidity by dropping the small particles (chips) into the analyzed river. Is this necessary, due to the natural turbidity of the rivers? I would like to hear the author's opinion.
-authors should avoid putting a lot of the references in one sentence, without negligible explanation or elaboration. For example, lines 36, 39, 40.
Author Response
Thanks for your time in reviewing the manuscript. Regarding your comments:
- Woodchips are added to the river for application of PIV to enable tracking also in case of absence of other trackable features (thus in case of absence of waves, foam, colour differences generated by sediment transport). As specified in the chapter about PIV “Only in one river (Guden Å), the water surface did not contain a sufficient number of natural particles for successful PIV analysis, thus artificial seeding (woodchips) was thrown into the water by two operators standing on an upstream bridge. In all surveyed XSs in Sweden, the streams showed a sufficient number of natural particles on the water surface (small floating wood particles, debris and especially foam generated by organic decomposition occurring in the nearby forests) and thus no artificial seeding was added.” In this regard, we think that the paper contains is innovative because we able to use PIV without artificial particles in 3 out of 4 rivers, significantly reducing the time typically used by operators to add artificial particles in the river.
- Furthermore, we have divided those citations groups into single citations to provide additional explanation.
Best Regards.
Round 2
Reviewer 4 Report
Dear Authors,
many thanks for having addressed my comments. I am satisfied with the present version.
This manuscript is a resubmission of an earlier submission. The following is a list of the peer review reports and author responses from that submission.
Round 1
Reviewer 1 Report
This manuscript presents an interesting attempt to determine flow velocities froma UAV using two techniqes, SVR and PIV. There are many remaining questions left after reading the manuscript. Some are listed in the annotated pdf, but a summary is provided here.
- The sensors are used mostly as black boxes with no detail on the algorithms used to filter data as well as questionable parameters, e.g. std of GNSS location is 0.2 mm for a hovering UAV.
- The methods to integrate all the measurements fro ma moving platform fail as uncertainties are not propagated which renders the velocity estimates useless.
- The validation with the ADCP compares surface with 20cm depth velocities, they could differ naturally or not, but compairson has no meaning if they could differ.
- The results do not show any sign of significance, the Std bands of the two methods are not even overlapping with eachother, nor the validation data.
- The manuscript would have been interesting to learn if those methods could actually succeed, but the manuscript simply accepts the given instruments with their parameters as listed by the manufacturer and this prevents the study from being successful.
I have doubts that the data collected can provide any significance due to the many parameters involved, which do not have a proper uncertainty assessment attached. Those which have are often missing critical parameters, e.g. what sampling frequency, over what time period or distance. The std of the GNSS on a UAV of 0.2 mm is already sufficient to disqualify the manuscript from publication. Neither can the UAV remain in such an accurate hovering position, nor will we ever know from a GNSS set-up.
For the reasons above, I recommend to reject the manuscript.

Reviewer 2 Report
Dear Authors,
the paper presents research comparing two methods of surface water velocity a case study of lowland European rivers. The topic can be considered interesting and original. Field measurements data (collected by PVI and UAS method) were calibrated with ground-truth velocity observations (ADCP). Research has a potential practical application on rivers with a lack of water flow monitoring network stations. In my opinion, the manuscript is written very well, congratulations.
I add some comments on different lines of the document.
Please confirm that UAS measurements and Ground-truth velocity observations (ADCP) were conducted at the same time?
Please explain how differences of “ADCP does not measure velocity at the water surface, but only 0.2 cm below it – line 372” influence on presented results?
Please provide information on how differences of morphological conditions of rivers on results?
Test conducted on Guden-Svostrupvej profile (L386) in my opinion is really bad located, you have been directly below the bridge – what determines many impacts of flow water, etc. additional water damming, spur from pillars, etc. also on the picture is shown that close to left bank is a lot of water plant, also indicated on water flow; also I see some no-flow parts – what makes difficult to water flow estimate.
--------------------------------------------
L1: “(Article, Review, Communication, etc.)” – Dear authors – please choose
L25: “UAS; PIV” – please add as the first full description
L55: “m/s” please use SI unit -> m s-1 and rest part of the manuscript (ms)
L77: “NASA” – if you use it the first time, please provide the full name of the institution
L89: “Fulton et al.” please provide a reference, year?
L91: “USA” as above
L99: I suggest add also “(v) morphological conditions of river channel”
L101-113 – please edit text
L226: Table 1: I suggest change coordinates of markers on the left and right bank change to one point of the center of the profile, and additional information about the width of river profile, average depth, etc. – it helps better-understood differences of investigated channels
L240: “few centimeters” please provide precise value?
Please add a location map with measurement stations.
Figure 4-7 part b) – please change the width of line – smaller please, also on part c) please add north arrow (direction)
Kind regards